# One-pot synthesis of hyperbranched polymers via visible light regulated switchable catalysis

Shuaishuai Zhu[1,2], Maoji Zhao[1,2], Hongru Zhou[1], Yingfeng Wen[1], Yong Wang [1] ✉, Yonggui Liao[1], Xingping Zhou[1] & Xiaolin Xie[1]

Switchable catalysis promises exceptional efficiency in synthesizing polymers with ever-increasing structural complexity. However, current achievements in such attempts are limited to constructing linear block copolymers. Here we report a visible light regulated switchable catalytic system capable of synthesizing hyperbranched polymers in a one-pot/two-stage procedure with commercial glycidyl acrylate (GA) as a heterofunctional monomer. Using (salen) Co[III]Cl (**1**) as the catalyst, the ring-opening reaction under a carbon monoxide atmosphere occurs with high regioselectivity (>99% at the methylene position), providing an alkoxycarbonyl cobalt acrylate intermediate (**2a**) during the first stage. Upon exposure to light, the reaction enters the second stage, wherein **2a** serves as a polymerizable initiator for organometallic-mediated radical self-condensing vinyl polymerization (OMR-SCVP). Given the organocobalt chain-end functionality of the resulting hyperbranched poly(glycidyl acrylate) (*hb*-PGA), a further chain extension process gives access to a core-shell copolymer with brush-on-hyperbranched arm architecture. Notably, the post-modification with 2,2,6,6-tetramethylpiperidine-1-oxyl (TEMPO) affords a metal-free *hb*-PGA that simultaneously improves the toughness and glass transition temperature of epoxy thermosets, while maintaining their storage modulus.

Diverse functions of biopolymers originate in large measure from their elegantly programmed sequence and architecture, demonstrating fascinating structure–property relationships that greatly inspire the development of advanced polymeric materials[1–5]. Historically, judiciously timed monomer additions, carefully predetermined monomer reactivity ratios, together with living polymerization methodologies have enabled access to a wealth of well-defined microstructures (block, cyclic, branched, etc.)[6–8]. However, these synthetic methodologies typically involve multiple-step procedures, orthogonal catalysis, and laborious operations, and more importantly, become impractical as the level of polymer complexity increases. A promising solution to these synthetic challenges may lie within the rapidly evolving field of switchable catalysis, wherein a single catalyst connects and discriminates different transformation cycles in a single efficient process[9–11]. It is envisioned that such a nascent catalytic strategy not only endows exceptional efficiency for synthesizing polymeric materials with the desired level of complexity but also circumvents laborious intermediate purification and minimizes catalyst usage[12,13].

In 2015, Williams et al. pioneered a switchable catalytic system involving ring-opening copolymerization (ROCOP) of epoxide/cyclic anhydride and ring-opening polymerization (ROP) of cyclic ester, which enables the straightforward synthesis of polyester-*b*-polyesters with little tapering from mixtures of monomers[14–16]. With the development of a myriad of organometallic[17–20] and oganic[21–24]

[1]School of Chemistry and Chemical Engineering, Huazhong University of Science and Technology, 430074 Wuhan, China. [2]These authors contributed equally: Shuaishuai Zhu, Maoji Zhao. ✉e-mail: tcwy@mail.hust.edu.cn

catalysts and the ongoing expansion of monomer scope[25–27], this self-switchable polymerization technique has evolved into the most popular and powerful platform to construct oxygenated block copolymers directly from mixtures of monomers. Notably, the sequences, compositions, and molecular weights of the resulting oxygenated block copolymers can be easily and precisely modulated, thus allowing for the on-demand development of advanced polymeric materials in applications, such as thermoplastic elastomers (TPEs), pressure-sensitive adhesives (PSAs), and toughened plastics[28–31]. Yet, the limitation is that self-switchable polymerization systems typically iterate between two polymerization cycles within ROCOP and ROP and only give access to linear AB or ABA oxygenated block copolymers. On the other hand, living/controlled radical polymerizations are among the most versatile techniques to construct polymers with a wide array of functionality[32–34]. The integration of living/controlled radical polymerization with ring-opening (co) polymerization is intriguing, as the resulting block copolymers may display attractive phase separation behaviors and thus serve as high-value-added polymeric materials[35–37]. Our group first demonstrated the quantitative transformation between Co–O and Co–C bonds, and thereby developed switchable catalytic systems comprising ROCOP of epoxide/CO₂/cyclic anhydride and organometallic-mediated radical polymerization (OMRP) of vinyl monomers, which gave facile access to copolymers connecting polyacrylates or poly(vinyl acetate) with the oxygenate blocks (Fig. 1a)[38,39]. Very recently, we successfully synthesized cyclic CO₂-based polycarbonates and polyesters in a one-pot process via cobalt-mediated switchable catalytic transformation from ROCOP of epoxide/CO₂/cyclic anhydride to intramolecular radical cyclization (Fig. 1b)[40].

Due to the globular and dendritic architectures, hyperbranched polymers (HBPs) display unique structures and properties such as abundant functional groups, intramolecular cavities, low viscosity, and high solubility, showing great potential in applications associated with mass/energy transfer, dissipation, and distribution[41–47]. Owing to the merits of controlled growth and site-specific initiation, self-condensing vinyl polymerization (SCVP) methodologies are gaining increasing research attention, albeit the non-trivial synthetic efforts and poor chemical stability of inimers (polymerizable initiators)[48–52]. Typically, the orthogonal catalytic transformation of a heterofunctional monomer using living/controlled polymerization techniques results in chemically crosslinked networks[53–55]. Yet, we rationalized that a switchable catalytic process involving the in situ synthesis of inimer and a subsequent SCVP would allow the one-pot synthesis of hyperbranched polymers from readily available heterofunctional feedstocks. In this vision, commercial glycidyl acrylate (GA) was judiciously chosen as a heterofunctional monomer, whereby an alkoxycarbonyl cobalt acrylate intermediate could form via cobalt-catalyzed ring-opening reaction under a CO atmosphere. Upon irradiation of visible light, the organometallic-mediated radical self-condensing vinyl polymerization (OMR-SCVP) smoothly occurs, giving access to hyperbranched poly(glycidyl acrylate) with (salen)Co chain-end function linked via a Co-C bond [hb-PGA-Co^{III}(salen)] (Fig. 1c).

## Results and discussion

### Synthesis and characterization of the alkoxycarbonyl cobalt acrylate inimer

In our primary effort to accomplish the switch, the stoichiometric ring-opening of GA operated by (salen)Co^{III}Cl (complex **1**) under 1 atm CO

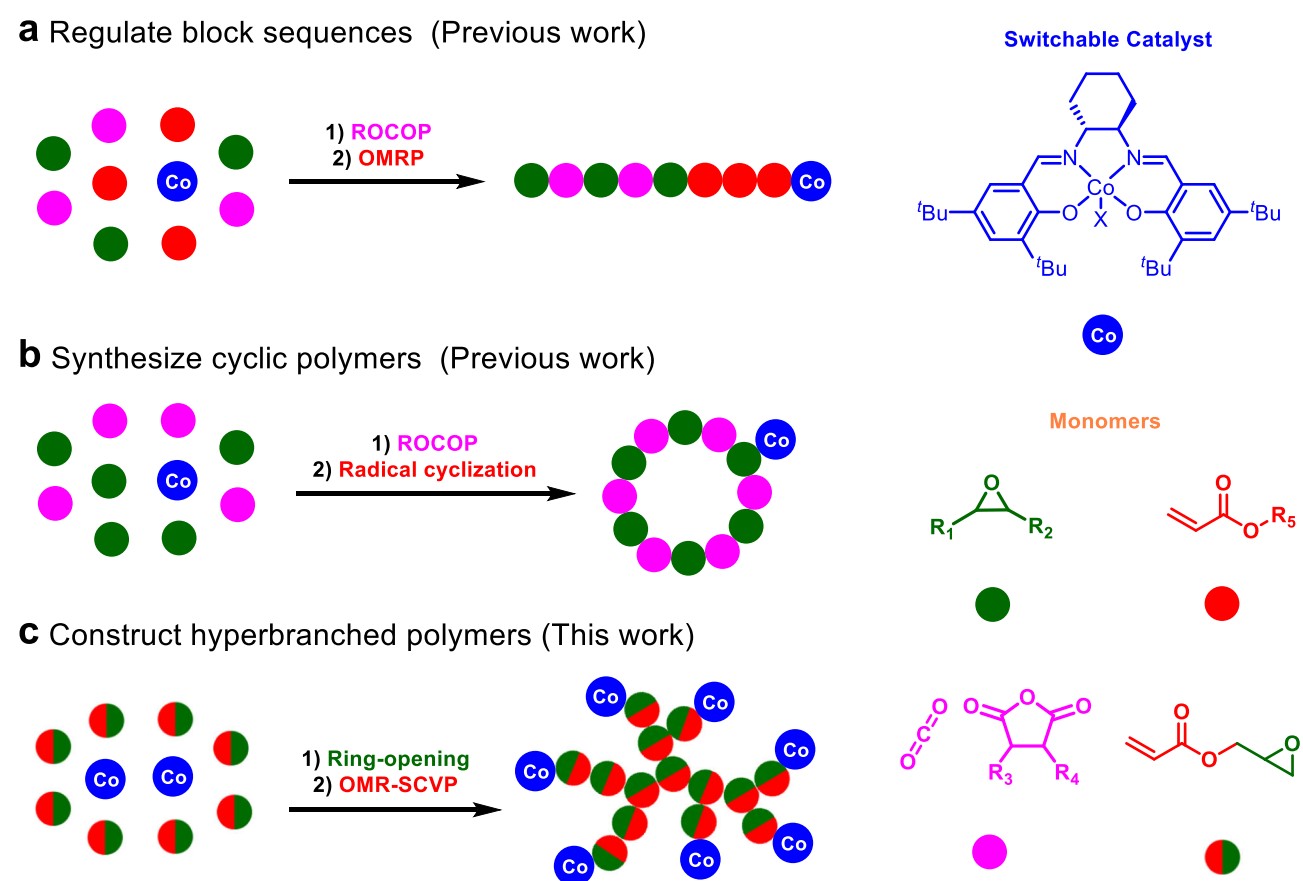

**Fig. 1 | One-pot synthesis of polymers with different topologies via cobalt-mediated switchable catalysis. a** Regulating block sequences from mixtures of epoxide, anhydride, and acrylate, X = Cl. **b** Synthesizing cyclic polyesters from mixtures of epoxide and anhydride, X = pentenoate. **c** Constructing hyperbranched polymers with GA as a common monomer, X = Cl.

was conducted in the dark at room temperature, aiming to obtain the inimers via migratory insertion of CO into the Co–O bond of the (salen) Co-alkoxo intermediates[38]. The reaction mixture was allowed to be stirred for 8 h before removing the excess GA by evaporation under a vacuum. The electrospray ionization mass spectrum (ESI-MS) of the resulting dark green powder displayed only two series of signals in accordance with [(salen)Co$^{III}$Cl + GA + CO]$^+$ and [(salen)Co$^{III}$Cl + GA + CO + Na]$^+$, respectively (Supplementary Fig. 1), verifying the quantitative transformation of **1** into the targeted alkoxycarbonyl cobalt acrylate species. Typically, the ring-opening of a mono-substituted epoxide via C–O cleavage can occur at either the methylene or methine carbon[56,57]. However, only one population of peaks ascribed to ester protons was observed approximately at 5.3 ppm on the $^1$H NMR spectrum (Supplementary Fig. 2). According to its relative integration area (one proton), the ester proton could be reasonably assigned to (salen)Co$^{III}$–CO$_2$C*H*– rather than (salen)Co$^{III}$–CO$_2$C*H*$_2$–. Moreover, the $^1$H COSY spectrum disclosed that the corresponding carbon connects to two methylene carbons, further verifying the sole formation of **2a** instead of **2b** (Supplementary Fig. 4). Therefore, it could be assumed that the nucleophilic attack of the axial Cl within **1** occurs at the less sterically hindered methylene carbon with >99% regioselectivity in the ring-opening reaction (Fig. 2).

## Synthesis of hyperbranched poly(glycidyl acrylate) via OMR-SCVP

With the highly selective synthesis of inimer, we sought to conduct the OMR-SCVP with GA ([GA]/[**2a**] = 30/1, [GA] = 2 M in THF) at room temperature under white light irradiation (household LED with a light intensity of 10 mW cm$^{-2}$). As revealed by $^1$H NMR spectroscopy, the conversion of GA reached 66% after exposure to light for 64 h. The number average molecular weight ($M_{n, SEC}$) and weight average molecular weight ($M_{w, SEC}$) of the resulting polymer was calculated as 9.4 and 14.2 kDa, respectively. The monomodal and broad distribution (Đ = 1.52) is ascribed to the well-controlled chain propagation but ununiformed branching process (Fig. 3a). Moreover, SEC-MALLS (multi-angle laser light scattering) was applied to determine the absolute molecular weights. Accordingly, the value (31.6 kDa) of $M_{n, MALLS}$ was significantly higher than that of $M_{n, SEC}$, further verifying the formation of a hyperbranched poly(glycidyl acrylate) [*hb*-PGA-Co$^{III}$(salen)]. For comparison, a linear PGA [*l*-PGA-Co$^{III}$(salen), $M_{n, SEC}$ = 7.0 kDa, Đ = 1.18] was synthesized (Supplementary Note 3)[38]. In this case, the values of $M_{n, SEC}$, $M_{w, SEC}$, $M_{n, MALLS}$ were calculated as 7.0, 8.3, and 9.9 kDa, respectively, which are very close to each other.

A piece of more direct evidence for the hyperbranched structure of the products came from their Mark–Houwink plots that were recorded by the SEC with a viscosity detector (Fig. 3b). Accordingly, the Mark–Houwink exponents (α) of *hb*-PGA-Co$^{III}$(salen) and *l*-PGA-Co$^{III}$(salen) were calculated as 0.35 and 0.75, respectively, which are identical for a randomly branched and linear polymer[52]. The ratio of the intrinsic viscosities of *hb*-PGA-Co$^{III}$(salen) and *l*-PGA-Co$^{III}$(salen), the contraction factor (g′), decreases with increasing molecular weights, also typical for hyperbranched polymers[50].

Encouraged by these experimental results, the cobalt-mediated switchable catalytic transformation of GA was further exploited at room temperature ([GA]/[**1**] = 30/1, [GA] = 2 M in THF), aiming to synthesize *hb*-PGA-Co$^{III}$(salen) in a single efficiently procedure (Table 1). The whole process comprises two stages: (1) ring-opening reaction under 1 atm CO in the dark; (2) OMR-SCVP regulated by visible light. As revealed by $^1$H NMR spectroscopy, the conversion of **1** was above 99% after reacting for 8 h in an ampule wrapped in aluminum foil. To ensure the completion of the ring-opening reaction, the mixture was allowed to be stirred at room temperature for another 4 h before exposure to white light LED (with a light intensity of 10 mW cm$^{-2}$). During the OMR-SCVP stage, the reaction was monitored by $^1$H NMR analysis of aliquots

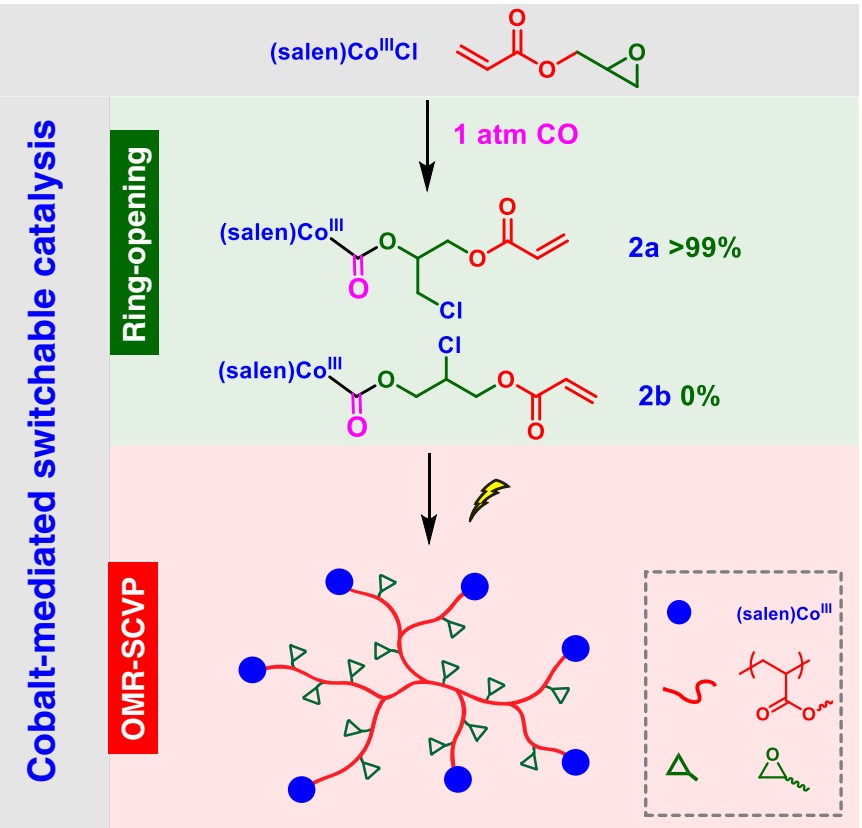

**Fig. 2 | Reaction diagram.** Visible light-regulated switchable catalysis of cobalt for one-pot synthesis of hyperbranched polymers.

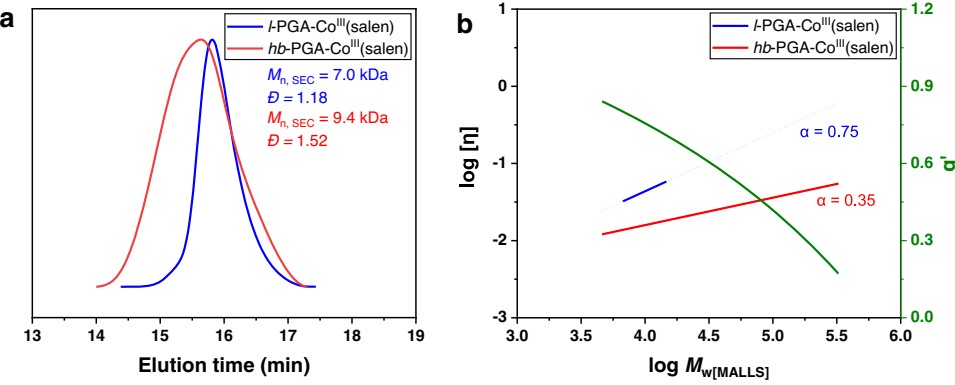

**Fig. 3 | The comparison between linear and hyperbranched PGA. a** SEC curves of *hb*-PGA-Co$^{III}$(salen) (red line) and *l*-PGA-Co$^{III}$(salen) (blue line). **b** Mark−Houwink plots for *hb*-PGA-Co$^{III}$(salen) (red line, $\alpha$ = 0.35) and *l*-PGA-Co$^{III}$(salen) (blue line, $\alpha$ = 0.75). Contraction factor, $g' = [\eta]_{br}/[\eta]_{lin}$ (green line).

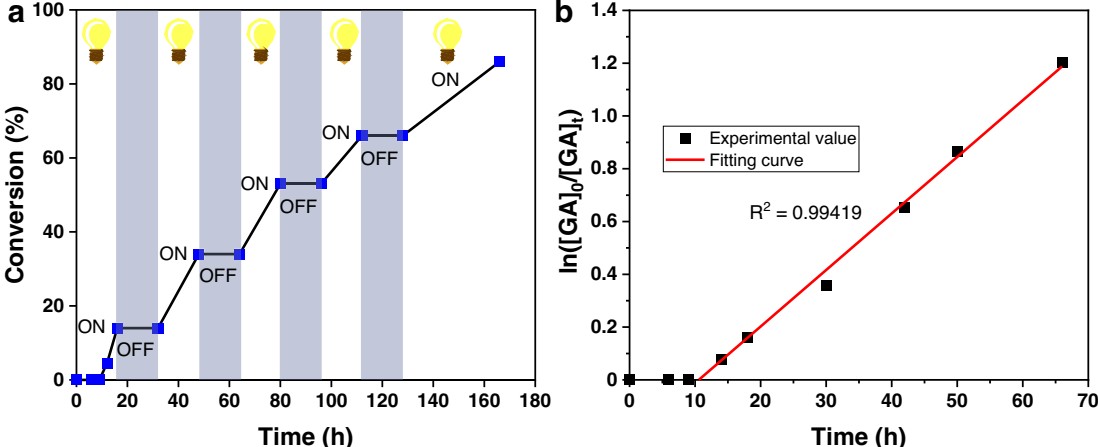

**Fig. 4 | Kinetic study of OMR-SCVP (household LED, 10 mW cm$^{-2}$), [GA]/[1] = 30/1, at room temperature under white light. a** Investigation of the "ON/OFF" switching by light. **b** Plots of ln([GA]$_0$/[GA]$_t$) vs. time.

taken at regular intervals. The polymerization smoothly proceeded, and GA conversion reached 14% after reacting for 16 h. However, no polymerization was observed during the following 16 h without white light irradiation, suggesting that a true ON/OFF living photopolymerization system could be developed. Exposure to light for a second 16 h turned the polymerization back on and this ON/OFF cycle can be repeated numerous times without observable reaction in the absence of visible light (Fig. 4a and Supplementary Fig. 5). Moreover, OMR-SCVP of GA was observed to proceed with linear first-order kinetics with an induction period of about 10 h and an apparent polymerization rate of $2.1 \times 10^{-2}$ h$^{-1}$ (Fig. 4b and Supplementary Fig. 6). When screening the polymerization under different light intensities, a positive correlation between the polymerization rate and light intensity could be observed. The conversions of GA are 28% and 48% when decreasing the light intensity to 3 and 5 mW cm$^{-2}$, respectively (Table 1, entries 4−5 and Supplementary Fig. 8). Moreover, the induction periods under the light density of 3 and 5 mW cm$^{-2}$ are 24 and 16 h, respectively, indicating that stronger light intensity is favorable for shortening the induction period.

When prolonging the reaction time, the conversion of GA, the $\alpha$ value, and $M_n$ of the products steadily increased (Table 1, entries 1−3), suggesting that the branching propagation mainly occurs at the early stage of OMR-SCVP. Moreover, the variation of $\alpha$ value verifies that the branching density of the polymeric products decreases with [GA]/[**1**] feed ratio. As is well known, the reaction rate of living/controlled radical polymerization is positively related to the concentration of radicals[58–61]. Interestingly, the conversions of GA under the [GA]/[**1**]

feed ratio of 10/1, 20/1, 50/1, 100/1, 500/1, and 1000/1 are 69% (96 h), 72% (96 h), 73% (66 h), 84% (66 h), 31% (66 h), and 16% (66 h), respectively, suggesting that the polymerization rates first increase and then decrease when decreasing the concentration of **1** (Table 1, entries 7−12 and Supplementary Fig. 11). Considering that visible light is essential for the homolytic cleavage of Co−C bonds, we envisioned that a concentrated solution of **2a** ([GA]/[**2a**] < 100/1, dark red) hinders the interaction between **2a** and light such that not all of **2a** can be converted to carbon-centered radicals. However, in the case of a dilute solution ([GA]/[**2a**] ≥ 100/1, pale red), light can easily penetrate through the solution, which facilitates the homolysis of Co−C bonds. In this context, the concentration of the radical active species first increases and then decreases with the decrement of the concentration of **2a** under the experimental conditions, which is in line with the tendency of the polymerization rate.

Furthermore, plausible mechanistic aspects that concern the one-pot synthesis of hyperbranched polymers via cobalt-mediated switchable catalytic transformation are proposed (Fig. 5). During the first stage, (salen)Co$^{III}$Cl effectively activates the epoxide moiety of GA via coordination, facilitates the nucleophilic attack of the axial Cl at the less sterically hindered methylene carbon, and enables the ring-opening of the epoxide moiety. The resulting (salen)Co-alkoxo intermediates then undergo a carbonylation reaction to generate the targeted alkoxycarbonyl cobalt acrylate species (**2a**) via migratory insertion of CO into the Co−O bonds. Upon exposure to visible light, the reaction enters the second stage, wherein the homolytic cleavage of the Co−C bonds of **2a**

results in the formation of acrylate acyl radicals that can both initiate the radical polymerization of GA via linear pathway and participate in the radical polymerization as monomers to create branches.

As the resulting *hb*-PGA-Co$^{III}$(salen) possesses (salen)Co chain-end functionality linked via Co–C bonds, they could serve as a macroinitiator for visible light-regulated OMRP of vinyl monomers. In this regard, the chain extension reaction of *hb*-PGA-Co$^{III}$(salen) ($M_{n,SEC}$ = 9.6 kDa, Đ = 1.78) with methyl acrylate (MA) was conducted in THF at room temperature, which enables the facile construction of a core−shell copolymer with brush-on-hyperbranched arm architecture [*hb*-PGA-*g*-PMA-Co$^{III}$(salen)] (Fig. 6a). According to the $^1$H NMR spectrum of the reaction mixture, the conversion of MA reached 35% was achieved after exposure to light for 36 h (Supplementary Fig. 18). The shift of the SEC curve toward a higher molecular weight (from 9.6 to 17.6 kDa) could verify the successful chain extension of MA along the polymer chain end of *hb*-PGA. Moreover, the decrease in the molecular weight distribution could be ascribed to the well-controlled manner in the linear chain propagation process (Fig. 6b).

## Post-functionalization and application of hyperbranched PGA

After the second stage of the whole procedure, excessive 2,2,6,6-tetramethylpiperidine-1-oxyl (TEMPO) was added to the reaction mixture as a radical scavenger, aiming at getting rid of the terminated organocobalt moieties (Table 1, entry 6). On the $^1$H NMR spectrum of the pale red and viscous solid product, characteristic signals of (salen)Co species disappeared, and new peaks corresponding to TEMPO could be observed, suggesting the successful synthesis of metal-free *hb*-PGA-TEMPO (Supplementary Fig. 20). Epoxy resins (EPs) represent the most widely used as matrix resins of high-performance composites in application scopes such as semiconductor packaging and electrical insulation[62]. However, they typically suffer from severe volume shrinkage during the curing processes and display low impact resistance. Hyperbranched epoxy modifiers can effectively toughen epoxy thermosets, yet, an obvious decrease in glass transition temperature ($T_g$) is always accompanied[63–66]. Moreover, hyperbranched epoxy compounds are typically synthesized via glycidylation of hyperbranched precursors, and epoxy groups are only located at the chain ends[67]. Given its high density of epoxy groups pendent along the flexible main chain, we envisioned that *hb*-PGA-TEMPO would provide a unique opportunity to fine-tune the thermal/mechanical performances of epoxy thermosets. In this regard, diglycidyl ether of bisphenol F epoxy resin (DGEBF, YDF-170) and *hb*-PGA-TEMPO ($M_{n,SEC}$ = 12.0 kDa, Đ = 2.06) were utilized as the matrix, and modifier respectively to prepare the *hb*-PGA-TEMPO/DGEBF blends (Fig. 7a). In the blends, the weight ratios of *hb*-PGA-TEMPO to DGEBF were fixed at 0/100, 4/96, and 8/92. The impact strength was measured to evaluate the toughness of neat and modified epoxy thermosets. As shown in Fig. 7b, adding 4 wt% *hb*-PGA-TEMPO increases the impact strength of the cured epoxy resin from 1.92 to 2.35 kJ/m$^2$, displaying a remarkable increment of 22%. Further increasing the *hb*-PGA-TEMPO content to 8 wt% led to a reduction in the impact strength of the blends owing to much increased cross-linking density. It should be noted that the impact strength of the blends is still comparable with that of the neat DGEBF. Moreover, the effect of *hb*-PGA-TEMPO on the thermal/mechanical properties of *hb*-PGA-TEMPO/DGEBF blends was investigated via DMA measurements. To our delight, while most hyperbranched epoxy modifiers toughen epoxy resins at the expense of their thermal properties[68], the presence of *hb*-PGA-TEMPO significantly increases the $T_g$s of cured *hb*-PGA-TEMPO/DGEBF blends.

**Table 1 | One-pot synthesis of *hb*-PGA-Co$^{III}$(salen) via visible light toggled switchable catalysis from the ring-opening reaction to OMR-SCVP**[a]

| Entry | [GA]/[1] | t (h) | Conv.[b] (GA, %) | $M_{n,SEC}$[c] (kDa) | Đ[c] | $M_{n,MALLS}$[d] (kDa) | α[e] |
|---|---|---|---|---|---|---|---|
| 1 | 30/1 | 43 | 41 | 5.8 | 1.46 | 15.7 | 0.27 |
| 2 | 30/1 | 50 | 53 | 8.6 | 1.59 | 23.5 | 0.31 |
| 3 | 30/1 | 65 | 70 | 9.6 | 1.78 | 35.7 | 0.38 |
| 4[f] | 30/1 | 65 | 48 | 6.9 | 1.70 | 17.5 | 0.28 |
| 5[g] | 30/1 | 65 | 28 | 3.9 | 1.31 | 14.9 | 0.25 |
| 6[h] | 30/1 | 84 | 79 | 12.0 | 2.06 | 41.8 | 0.36 |
| 7 | 10/1 | 96 | 69 | 4.5 | 1.31 | 16.6 | 0.24 |
| 8 | 20/1 | 96 | 72 | 7.1 | 1.74 | 19.3 | 0.28 |
| 9 | 50/1 | 66 | 73 | 16.2 | 2.38 | 47.1 | 0.39 |
| 10 | 100/1 | 66 | 84 | 19.6 | 2.23 | 53.6 | 0.43 |
| 11 | 500/1 | 66 | 31 | 34.7 | 2.09 | 61.0 | 0.44 |
| 12 | 1000/1 | 66 | 16 | 31.1 | 1.91 | 76.5 | 0.46 |

[a]Conditions: room temperature; [GA] = 2 M in THF; ring-opening reaction run for 12 h; light source for OMR-SCVP: household white LED lamp (10 mW cm$^{-2}$ at the sample position).
[b]Determined by $^1$H NMR spectroscopy.
[c]Determined by SEC analysis in THF calibrated against polystyrene standards.
[d]SEC/MALLS.
[e]Mark–Houwink exponent, SEC/viscosity in THF.
[f]Exposure to white light LED with a light intensity of 5 mW cm$^{-2}$.
[g]Exposure to white light LED with a light intensity of 3 mW cm$^{-2}$.
[h]Terminated with TEMPO.

**Fig. 5 | Plausible mechanistic aspects.** The one-pot synthesis of hyperbranched polymers via cobalt-mediated switchable catalytic transformation of GA.

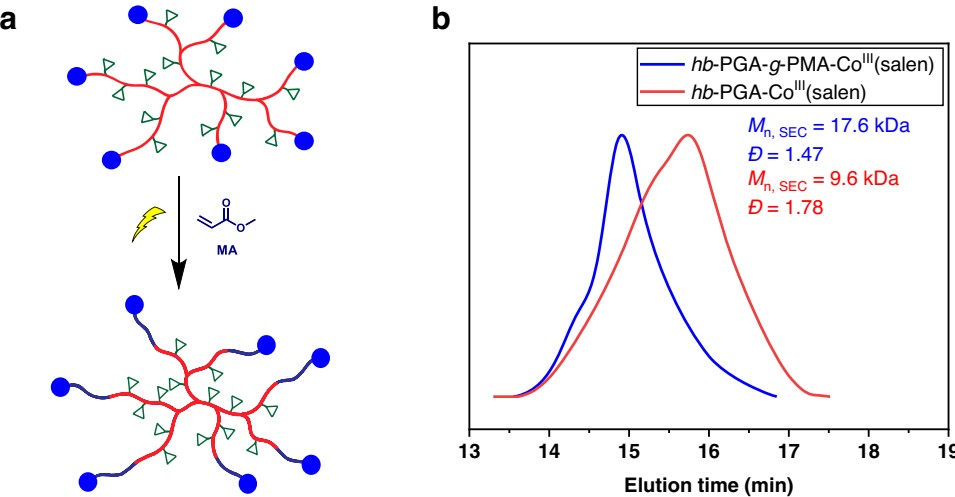

**Fig. 6 | Chain extension of hyperbranched PGA. a** Chain extension of *hb*-PGA-Co^III(salen) with MA to synthesize *hb*-PGA-*g*-PMA-Co^III(salen). **b** SEC traces of *hb*-PGA-Co^III(salen) (red line) and *hb*-PGA-*g*-PMA-Co^III(salen) (blue line).

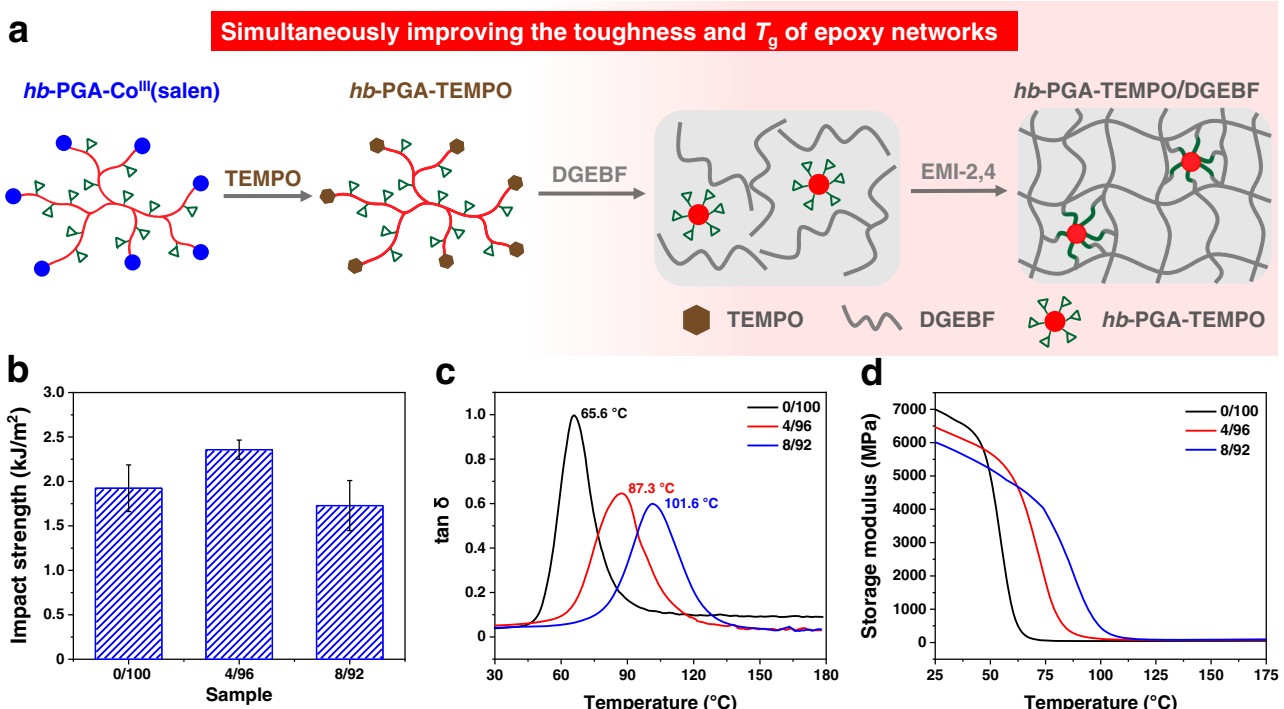

**Fig. 7 | Post-functionalization of hyperbranched PGA. a** Schematic illustration for synthesizing *hb*-PGA-TEMPO via post-functionalization of *hb*-PGA-Co^III(salen) with TEMPO, and preparing *hb*-PGA-TEMPO/DGEBF blends. **b** Impact strength, **c** $T_g$, and **d** Storage modulus of *hb*-PGA-TEMPO/DGEBF blends with different weight ratios. The error bar represents the standard deviation of three measurements.

From the DMA results, the $T_g$s of cured DGEBF, 4 wt% *hb*-PGA-TEMPO/DGEBF, and 8 wt% *hb*-PGA-TEMPO/DGEBF were 66, 87, and 102 °C, respectively, which is consistent with those obtained from DSC analysis. (Fig. 7c and Supplementary Fig. 22).

Moreover, the cross-linking densities of the cured DGEBF and the blends were calculated based on rubber elasticity theory[69]. Accordingly, the cross-linking density increases when increasing the content of *hb*-PGA-TEMPO, which could help rationalize the trend of $T_g$ (Supplementary Table 1). According to Fig. 7d, the storage modulus of the as-prepared *hb*-PGA-TEMPO/DGEBF blends is only slightly lower than that of the neat DGEBF.

In conclusion, the first example of a switchable catalytic system capable of constructing hyperbranched polymers in a single efficient

procedure was demonstrated. Using visible light as an external stimulus, (salen)Co^IIICl bridged and discriminated two catalytic transformations with GA as a heterofunctional substrate: (1) highly regioselective ring-opening reaction under CO that allows in situ formation of an alkoxycarbonyl cobalt acrylate intermediate as the inimer and (2) organometallic-mediated radical self-condensing vinyl polymerization. The kinetic study demonstrated the successful development of a true ON/OFF living polymerization system, whereby the chain propagation process can be regulated by visible light and the branching structure of the products can be easily tailored and modified. By varying the feed ratio of [monomer]:[cobalt], the branching structure of the products can be easily tailored and modified. Notably, the organocobalt chain-end functionality permits a further chain

extension reaction that gives facile access to core–shell copolymers with brush-on-hyperbranched arm architecture. Notably, the post-modification with TEMPO afforded a metal-free hyperbranched epoxy polymer that could simultaneously improve the toughness and $T_g$ of epoxy thermosets, while maintaining their storage moduli. The present study opens an avenue to construct polymers with a high degree of complexity with exceptional efficiency by taking advantage of switchable catalysis. Given that transition metal catalysis rises concerns with respect to sustainability and renders the cobalt-mediated switchable catalytic strategy less attractive for larger-scale application, our future efforts will be directed toward the development of metal-free Lewis pair catalytic systems capable of integrating ROCOP or ROP with living/controlled radical polymerization for the one-pot synthesis of advanced polymeric materials with a high degree of structural complexity.

## Methods

### OMR-SCVP of GA with 2a as the inimer

In an argon-filled glove box, **2a** (79.4 mg, 0.1 mmol), GA (0.34 mL, 3 mmol), and THF (1.16 mL) were charged into a 10 mL ampoule equipped with a magnetic stir bar. The ampoule was then taken out of the glove box and allowed to be stirred at 25 °C under irradiation of a household white LED lamp with a light intensity of 10 mW cm$^{-2}$. A small aliquot of the polymerization mixture was taken out for $^1$H NMR spectroscopy and the remained crude mixture was precipitated in cold methanol.

### Visible light-regulated switchable catalysis from ring-opening to OMR-SCVP with GA as a heterofunctional monomer

In an argon-filled glove box, (salen)Co$^{III}$Cl complex (0.64 g, 1 mmol), GA (3.44 mL, 30 mmol), and THF (11.56 mL) were charged into a 25 mL ampoule, which was wrapped in aluminum foil and equipped with a magnetic stir bar. The ampoule was then taken out of the glove box and allowed to be stirred under 1 atm CO at room temperature for 12 h. Next, the aluminum foil was uncovered and the ampoule was degassed and pressurized with 1 atm argon and allowed to be stirred at ambient temperature under the irradiation of a household white LED lamp with a light intensity of 10 mW cm$^{-2}$. In the kinetic study, aliquots were withdrawn by nitrogen-purged syringes from the reaction mixture at predetermined interval times and analyzed by $^1$H NMR to measure the conversions.

## Data availability

The authors declare that the data supporting this study are available within the paper and its supplementary information files. Any other data are available from the corresponding authors upon request.

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

## Acknowledgements

Thank the Financial support from the National Key R&D Plan (No. 2016YFB0302400), NSFC (No. 21604027) of China, as well as the analytical and testing assistance from the Analysis and Testing Center of HUST.

## Author contributions

Y. Wang conceived and designed the experiments, and revised the manuscript. S.Z., M.Z., and H.Z. conducted the experiments, analyzed the data and wrote the draft. Y. Wen, Y.L., X.Z., and X.X. discussed the experimental results and revised the manuscript.

## Competing interests

The authors declare no competing interests.
