## [Peer Review File · Nature Communications]

One-pot synthesis of hyperbranched polymers via visible light regulated switchable catalysisReviewers' Comments:

Reviewer #1:

Remarks to the Author:

The manuscript by Zhu et al. reported the One-Pot Synthesis of Hyperbranched Polymers via Visible Light Regulated Switchable Catalysis. Using (salen)CoIII as the catalyst, the ring-opening reaction under a carbon monoxide atmosphere occurs with high regioselectivity, providing an alkoxy carbonyl cobalt acrylate intermediate during the first stage. Upon exposure to light, the reaction enters the second stage, undergoing organometallic-mediated radical self-condensing vinyl polymerization). Overall, this work is complimentary to the authors' previous one (Angew. Chem. Int. Ed. 2020, 59, 5988-5994). Due to the lack of sufficient novelty, I don't think the paper fits well with the high scope of Nature Communications. The following are some specific questions for the authors to consider:

1. I am wondering about the accuracy of the M_n , SEC values since they are relative values to polystyrene standards.
2. The tailing and asymmetry of the SEC traces in Figure 3 concerns me about the actual utility of the core-shell copolymer designs and strategies to prepare hyperbranched polymers.

Reviewer #2:

Remarks to the Author:

This paper describes a new and promising method to synthesize hyperbranched polymers and related (mostly star-shaped) macromolecular structures based on photoswitchable catalysts. This is an interesting idea that has been carefully carried out by well-designed experiments which have been properly carried out, described and interpreted, justifying also the conclusions of the paper. The reference list is extensive and to the point. To the best of my knowledge, the described methodology has not yet been applied to hyperbranched structures and I do agree with the authors that it has quite generic potential in this particular field.

For the reasons above, I'm convinced that this work deserves publication without any major revisions. Working myself in the field of polymer chemistry, however, it is not easy for me to judge whether the generic impact and interest to a broad scientific audience lives up to the expectations of this particular journal. In this respect, I would like to suggest that if the paper would be accepted, some additional explanation about the catalytic mechanism could strongly improve the readability by a non-specialized audience. Moreover, I would appreciate if the authors could also mention that transition metal catalysis (and certainly also cobalt-based) rises concerns with respect to sustainability, rendering the strategy less attractive for larger scale application.

Reviewer #3:

Remarks to the Author:

This work reported a visible light regulated switchable catalytic system capable of synthesizing hyperbranched polymers in a one-pot/two-stage procedure. All experiments and characterizations were well performed and this approach proposes a powerful tool to achieve polymerization with a easily switchable on/off capability. I think the article can publish in nature comm after following questions answered.

1. I didn't find the degree of branching for the hyperbranch polymers synthesized in the manuscript. Is it possible to calculate DB of the product?
2. Only one type of light (household LED, light intensity of 10 mW·cm⁻²) was used in the manuscript. I think it is valuable to study the effect of different light intensities. The author also mentioned the

color of the solution would affect the light penetration, resulting in different polymerization rates. This speculation also needs further proof.

3. Too much misspelling. In Scheme 1, 'Previours work' should be 'Previous work', 'hyperbranced' should be 'hyperbranched', and 'OMR-SCVCP' should be 'OMR-SCVP' according to the manuscript. The author should take a carefully recheck.

Response to the reviewers' comments and description of changes made to the manuscript.

Reviewer #1

The manuscript by Zhu et al. reported the One-Pot Synthesis of Hyperbranched Polymers via Visible Light Regulated Switchable Catalysis. Using (salen)Co^{III} as the catalyst, the ring-opening reaction under a carbon monoxide atmosphere occurs with high regioselectivity, providing an alkoxy carbonyl cobalt acrylate intermediate during the first stage. Upon exposure to light, the reaction enters the second stage, undergoing organometallic-mediated radical self-condensing vinyl polymerization). Overall, this work is complimentary to the authors' previous one (*Angew. Chem. Int. Ed.* **2020**, *59*, 5988). Due to the lack of sufficient novelty, I don't think the paper fits well with the high scope of Nature Communications. The following are some specific questions for the authors to consider:

Response: Thank the reviewer for the precious time and effort devoted to this manuscript! Also, we are grateful for the insightful comments and good suggestions that help us improve the overall quality of this manuscript. However, we do not agree with the reviewer that this work is complementary to our previous one (*Angew. Chem. Int. Ed.* **2020**, *59*, 5988) for the following reasons:

Switchable polymerization is a rapidly evolving field that is drawing increasing research attention, and a lot of research work in this field has been published in high-impact journals just in the past two years (*Nat. Commun.***2022**, *13*, 163; *Nat. Commun.***2021**, *12*, 7124; *Nat. Commun.***2022**, *13*, 163; *J. Am. Chem. Soc.* **2022**, *144*, 19896; *J. Am. Chem. Soc.* **2022**, *144*, 17905; *J. Am. Chem. Soc.* **2022**, *144*, 20687; *J.*

Am. Chem. Soc. **2022**, *144*, 6882; *J. Am. Chem. Soc.* **2022**, *144*, 17477; *J. Am. Chem. Soc.* **2021**, *143*, 10021; *Angew. Chem. Int. Ed.* **2022**, *61*, e2021154; *Angew. Chem. Int. Ed.* **2022**, *61*, e2021175; *Angew. Chem. Int. Ed.* **2021**, *60*, 16974; *Angew. Chem. Int. Ed.* **2021**, *60*, 9274). **However, all these encouraging achievements only involve ring-opening copolymerization (ROCOP) and ring-opening polymerization (ROP), and give access to linear oxygenated block copolymers.** As the reviewer mentioned, our group reported the first example of a switchable polymerization system that iterates between ROCOP of epoxide/anhydride/CO₂ and organometallic mediated radical polymerization (OMRP) of vinyl monomers, affording block copolymers connecting polyacrylates or poly(vinyl acetate) with the oxygenate blocks (*Angew. Chem. Int. Ed.* **2020**, *59*, 5988). **Prof. C. K. Williams, the most outstanding researcher in the field of switchable polymerization, proposed that future switch catalysis must drive to access other polymerization mechanisms besides ROCOP and ROP in her latest review in JACS (*J. Am. Chem. Soc.* **2021**, *143*, 10021).** She thinks that the **development of catalysts switchable between metal-carbon and metal-oxygen intermediates holds great opportunities and took our work as the only example.** Moreover, the rapid and quantitative transformation from Co-O to Co-C bonds via migratory insertion of carbon monoxide (CO) **provides a straightforward and environmental-benign approach to the important alkoxycarbonyl cobalt complexes** (*Chem. Rev.* **2019**, *119*, 6906). Although CO insertions into metal-carbon single and multiple bonds are implicated in many stoichiometric and catalytic transformations of great significance in both industrial and academic fields, those into

a metal-oxygen bond are rare and only related to the Group 10 metals, and all concern metals in oxidation states up to II. Given that carbon-centered radicals are among the most important and common intermediates in both organic chemistry and polymer science, we envision that switchable catalytic systems based on the transformation between Co-O and Co-C bonds hold great promise in tackling the challenge to synthesize topological polymers with a high degree of structural complexity in a one-pot procedure. **In this manuscript, we successfully push the limit of switchable polymerization from regulating block sequences to constructing hyperbranched polymers.** Moreover, we demonstrate that the novel and unique hyperbranched polymeric product, with a flexible main chain and high-density pendant epoxy group, **can simultaneously improve the toughness and glass transition temperature of epoxy thermosets while maintaining their storage modulus.** Epoxy resins (EPs) are the most widely used thermosetting materials with broad application scopes, yet, compared with conventional thermosetting resins, they suffer from a reduction in volume shrinkage during the curing process and display poor resistance for crack initiation/propagation. In this regard, **we think this work shall attract a broad scientific audience in the fields of both polymer chemistry and materials science.**

1. I am wondering about the accuracy of the $M_{n, SEC}$ values since they are relative values to polystyrene standards.

Response: We agree with the reviewer that the accuracy of the $M_{n, SEC}$ values is questionable, since they are relative values to polystyrene standards. However, SEC is

still the most widely used method for determining the molecular weight of polymers, because there is none method that can outperform SEC in all aspects. In this regard, we beg for your understanding. In this work, we also conducted SEC-MALLS (multi-angle laser light scattering) analysis to determine the absolute molecular weights. Moreover, using SEC with a viscosity detector allows the recording of Mark-Houwink plots. The molecular weights obtained from the three methods can well elucidate the hyperbranched structure of the polymeric products obtained from the switchable polymerization.

2. The tailing and asymmetry of the SEC traces in Figure 3 concerns me about the actual utility of the core-shell copolymer designs and strategies to prepare hyperbranched polymers.

Response: We quite understand the concerns of the reviewer. In fact, tailing and asymmetry of the SEC traces are very common for polymeric products obtained from the SCVP methods, which is ascribed to the uncontrollable branching. In fact, the polydispersity distribution of the hyperbranched polymeric products obtained in this manuscript by visible light regulated OMR-SCVP is much narrower than those in the literature (See: *Angew. Chem. Int. Ed.* **2022**, *61*, e202211713; *Nat. Commun.* **2017**, *8*, 1863; *Macromolecules* **2017**, *50*, 9115-9120; *Macromolecules* **2013**, *46*, 6751-6757; *Macromolecules* **2011**, *44*, 2034–2049; *Macromolecules* **2008**, *41*, 7368-7373).

Reviewer #2

This paper describes a new and promising method to synthesize hyperbranched polymers and related (mostly star-shaped) macromolecular structures based on photo switchable catalysts. This is an interesting idea that has been carefully carried out by well-designed experiments which have been properly carried out, described and

interpreted, justifying also the conclusions of the paper. The reference list is extensive and to the point. To the best of my knowledge, the described methodology has not yet been applied to hyperbranched structures and I do agree with the authors that it has quite generic potential in this particular field.

For the reasons above, I'm convinced that this work deserves publication without any major revisions. Working myself in the field of polymer chemistry, however, it is not easy for me to judge whether the generic impact and interest to a broad scientific audience lives up the expectations of this particular journal. In this respect, I would like to suggest that if the paper would be accepted, some additional explanation about the catalytic mechanism could strongly improve the readability by a non-specialized audience. Moreover, I would appreciate if the authors could also mention that transition metal catalysis (and certainly also cobalt-based) rises concerns with respect to sustainability, rendering the strategy less attractive for larger scale application.

Response: We thank the reviewer for the appreciation! Switchable polymerization is a rapidly evolving field that is drawing increasing research attention, and 13 research work in this field has been published in top journals just in the past two years (3 *Nat. Commun.*; 6 *J. Am. Chem. Soc.*; 4 *Angew. Chem. Int. Ed.*). However, all these encouraging achievements only involve ring-opening copolymerization (ROCOP) and ring-opening polymerization (ROP) and give access to linear oxygenated block copolymers. **In this manuscript, we successfully push the limit of switchable polymerization from regulating block sequences to constructing hyperbranched polymers.** On the other hand, epoxy resins (EPs) are emerging as the most widely used thermosetting materials in semiconductor packaging materials, electrical insulation substances, and high-performance composites. Yet, compared with conventional thermosetting resins, they suffer from a reduction in volume shrinkage during the curing process and display poor resistance to crack initiation/propagation. Here we demonstrate that **the novel and unique hyperbranched polymeric product**, with a flexible main chain and high-density pendant epoxy group, **can simultaneously improve the toughness and glass transition temperature of epoxy thermosets while maintaining their storage modulus.** In this regard, **we think this work shall attract**

a broad scientific audience in the fields of both polymer chemistry and materials science.

Moreover, we are grateful for the insightful comments and good suggestions! Accordingly, we have added **Figure 5** in the revised manuscript concerning the plausible mechanistic aspects for the one-pot synthesis of hyperbranched polymers via cobalt-mediated switchable catalytic transformation of GA. Also, detailed discussions have been provided as: During the first stage, (salen)Co^{III}Cl effectively activates the epoxide moiety of GA via coordination, facilitates the nucleophilic attack of the axial Cl at the less sterically hindered methylene carbon, and enables the ring-opening of the epoxide moiety. The resulting (salen)Co-alkoxo intermediates then undergo a carbonylation reaction to generate the targeted alkoxy carbonyl cobalt acrylate species (**2a**) via migratory insertion of CO into the Co-O bonds. Upon exposure to visible light, the reaction enters the second stage, wherein the homolytic cleavage of the Co-C bonds of **2a** results in the formation of acrylate acyl radicals that can both initiate the radical polymerization of GA via linear pathway and participate in the radical polymerization as monomers to create branches.

Figure 5. Plausible mechanistic aspects for the one-pot synthesis of hyperbranched polymers via cobalt-mediated switchable catalytic transformation of GA.

Moreover, we quite agree with the reviewer that transition metal catalysis rises concerns with respect to sustainability, rendering the strategy less attractive for large-scale applications. In fact, we have already directed our efforts in developing metal-free

Lewis pair catalytic systems capable of integrating ROCOP with living/controlled radical polymerization for the one-pot synthesis of polymers with a high degree of structural complexity. Related discussions have been added to the conclusion part of the revised manuscript: Given that transition metal catalysis rises concerns with respect to sustainability and render the cobalt-mediated switchable catalytic strategy less attractive for larger scale application, our future efforts will be directed toward the development of metal-free Lewis pair catalytic systems capable of integrating ROCOP or ROP with living/controlled radical polymerization for the one-pot synthesis of advanced polymeric materials with a high degree of structural complexity.

Reviewer #3

This work reported a visible light regulated switchable catalytic system capable of synthesizing hyperbranched polymers in a one-pot/two-stage procedure. All experiments and characterizations were well performed and this approach proposes a powerful tool to achieve polymerization with a easily switchable on/off capability. I think the article can publish in nature comm after following questions answered.

Response: We thank the reviewer for the appreciation of our work, for the insightful comments and for the useful suggestions, which have helped improve the overall quality of the manuscript. We have now revised the manuscript and the supporting information accordingly.

1. I didn't find the degree of branching for the hyperbranch polymers synthesized in the manuscript. Is it possible to calculate DB of the product?

Response: We thank the reviewer for the comments. In fact, we attempted to calculate DB of the *hb*-PGA-Co^{III}(salen) by ¹H NMR spectroscopy. Unfortunately, the characteristic signals of the branching units completely overlap with those of epoxy group in *hb*-PGA-Co^{III}(salen). Therefore, DB of the hyperbranched polymer cannot be calculated accurately.

2. Only one type of light (household LED, light intensity of 10 mW·cm⁻²) was used in

the manuscript. I think it is valuable to study the effect of different light intensities. The author also mentioned the color of the solution would affect the light penetration, resulting in different polymerization rates. This speculation also needs further proof.

Response: Many thanks for the illuminating comments and good suggestions! Accordingly, we have further conducted two comparative experiments under a light intensity of 3 and 5 $\text{mW}\cdot\text{cm}^{-2}$, respectively. A positive correlation between the polymerization rate and light intensity could be observed. The conversions of GA are 28% and 48% when decreasing the light intensity to 3 $\text{mW}\cdot\text{cm}^{-2}$ and 5 $\text{mW}\cdot\text{cm}^{-2}$, respectively. Moreover, the induction periods under the light density of 3, 5, and 10 $\text{mW}\cdot\text{cm}^{-2}$ are 24 h, 16 h, and 10 h, respectively, indicating that stronger light intensity is favorable for shortening the induction period. The above highlighted discussion and the new experimental data have been provided in the revised manuscript (**Table 1**, entries 4-5) and the revised supplementary information (**Supplementary Figures 8-10**), respectively. Also, to confirm that polymerization rate is closely correlated to light penetration, we have further conducted two comparative experiments ([GA]/[**1**] feed ratio of 500/1 and 1000/1). When the concentration of **1** decrease, the color of the solution gradually changes from dark red to pale red, and the polymerization rate first increase and then decrease. Detailed discussions have been added as: As is well known, the reaction rate of living/controlled radical polymerization is positively related to the concentration of radicals. Interestingly, the conversions of GA under the [GA]/[**1**] feed ratio of 10/1, 20/1, 50/1, 100/1, 500/1, and 1000/1 are 69% (96 h), 72% (96 h), 73% (66 h), 84% (66 h), 31% (66 h), and 16% (66 h), respectively, suggesting that the polymerization rates first increase and then decrease when decreasing the concentration of **1** (**Table 1**, entries 7-12 and **Supplementary Fig. 11**). Considering that visible light is essential for the homolytic cleavage of Co-C bonds, we envisioned that a concentrated solution of **2a** ([GA]/[**2a**] < 100/1, dark red) hinders the interaction between **2a** and light such that not all of **2a** can be converted to carbon-centered radicals. However, in the case of a dilute solution ([GA]/[**2a**] > 100/1, pale red), light can easily penetrate through the solution, which facilitates the homolysis of Co-C bonds. In this context, the concentration of the radical active species first increases and then decreases

with the decrement of the concentration of **2a** under the experimental conditions, which is in line with the tendency of the polymerization rate. The new experimental data have been provided in the revised manuscript (Table 1, entries 11 and 12) and the revised supplementary information (Supplementary Figures 11, 16, 17), respectively.

3. Too much misspelling. In Scheme 1, 'Previous work' should be 'Previous work', 'hyperbranched' should be 'hyperbranched', and 'OMR-SCVCP' should be 'OMR-SCVP' according to the manuscript. The author should take a carefully recheck

Response: We are sorry for these avoidable mistakes and thank the reviewer for the carefulness! We have corrected these mistakes accordingly in the revised manuscript and carefully rechecked the full text. The new Figure 1 (as shown below) has been included in the revised manuscript.

Figure 1. One-pot synthesis of polymers with different topologies via cobalt-mediated switchable catalysis. a Regulating block sequences from mixtures of epoxide, anhydride, and acrylate, X = Cl. **b** Synthesizing cyclic polyesters from mixtures of epoxide and anhydride, X = pentenoate. **c** Constructing hyperbranched polymers with GA as a common monomer, X = Cl.

Reviewers' Comments:

Reviewer #1:

Remarks to the Author:

Although the points Wang and coauthors raise about the potential interest in the ability constructing hyperbranched polymers, I feel that the concerns regarding the degree of advance over previous work (Angew. Chem. Int. Ed. 2020, 59, 5988) are sufficiently important as to prohibit publication of this work in Nature Communications. I recognize that there are difference in the polymer structures or properties between this study and previous publications, however, I don't feel that these distinctions are such that the results reported are a significant leap forward.

Taken together, this work would be nicely served in "Macromolecules", where it would reach much of its target audience. This readership would not need as much "originality" as this reviewer or the wider Nature Communications readership.

Reviewer #2:

Remarks to the Author:

I'm satisfied wit the response of the authors and the modifications applied accordingly, to my first referee report (as well as those from the other reviewers to the extent I was able to judge this).

Hence, I could agree that this manuscript is published in Nature communications, provided the editors consider the novelty sufficiently high.

Reviewer #3:

Remarks to the Author:

Accept. The author answered all questions properly.

Reply to Reviewers' comments on Manuscript (NCOMMS-22-43038):

One-Pot Synthesis of Hyperbranched Polymers via Visible Light Regulated

Switchable Catalysis

Response to the reviewers' comments and description of changes made to the manuscript.

Reviewer #1

Although the points Wang and coauthors raise about the potential interest in the ability constructing hyperbranched polymers, I feel that the concerns regarding the degree of advance over previous work (Angew. Chem. Int. Ed. 2020, 59, 5988) are sufficiently important as to prohibit publication of this work in Nature Communications. I recognize that there are difference in the polymer structures or properties between this study and previous publications, however, I don't feel that these distinctions are such that the results reported are a significant leap forward.

Taken together, this work would be nicely served in "Macromolecules", where it would reach much of its target audience. This readership would not need as much "originality" as this reviewer or the wider Nature Communications readership.

Response: Thank the reviewer for the precious time and effort devoted to this manuscript! Also, we are grateful for the insightful comments and good suggestions that have helped us improve the overall quality of this manuscript. However, we do not quite agree with the reviewer about the novelty and the readership of this work.

A central focus in modern polymer chemistry is to precisely regulate polymer sequence and architecture for the development of advanced materials. However, traditional synthetic methodologies typically involve multiple-step procedures,

orthogonal catalysis, laborious operations, and more importantly, become impractical as the level of polymer complexity increases.

Regulated by stimuli, switchable polymerization bridges and discriminates distinct polymerization techniques, thus allowing facile synthesis of well-structured polymers in a single efficient process. In this regard, switchable polymerization is drawing increasing research attention, and a lot of research work in this field has been published in high-impact journals just in the past two years (*Nat. Commun.* **2022**, *13*, 163; *Nat. Commun.* **2021**, *12*, 7124; *J. Am. Chem. Soc.* **2022**, *144*, 19896; *J. Am. Chem. Soc.* **2022**, *144*, 17905; *J. Am. Chem. Soc.* **2022**, *144*, 20687; *J. Am. Chem. Soc.* **2022**, *144*, 6882; *J. Am. Chem. Soc.* **2022**, *144*, 17477; *J. Am. Chem. Soc.* **2021**, *143*, 10021; *Angew. Chem. Int. Ed.* **2022**, *61*, e2021154; *Angew. Chem. Int. Ed.* **2022**, *61*, e2021175; *Angew. Chem. Int. Ed.* **2021**, *60*, 16974; *Angew. Chem. Int. Ed.* **2021**, *60*, 9274). However, all these encouraging achievements only involve ring-opening copolymerization (ROCOP) and ring-opening polymerization (ROP), and give access to linear oxygenated block copolymers. As the reviewer mentioned, our group reported the first example of a switchable polymerization system that iterates between ROCOP of epoxide/anhydride/CO₂ and organometallic mediated radical polymerization (OMRP) of vinyl monomers, affording block copolymers connecting polyacrylates or poly(vinyl acetate) with the oxygenate blocks (*Angew. Chem. Int. Ed.* **2020**, *59*, 5988). **In the current manuscript, we successfully push the limit of switchable polymerization from regulating block sequences to constructing hyperbranched polymers.**

Therefore, we think it hold both significance and potential in the field of switchable polymerization. On the other hand, the resulting hyperbranched polymeric product, with a flexible main chain and high-density pendant epoxy group, can simultaneously improve the toughness and glass transition temperature of epoxy thermosets while maintaining their storage modulus, which is highly desirable yet rarely achieved for epoxy resins (EPs). In this regard, **we think this work shall attract a broad scientific audience in the fields of both polymer chemistry and materials science.**

Reviewer #2

I'm satisfied with the response of the authors and the modifications applied accordingly, to my first referee report (as well as those from the other reviewers to the extent I was able to judge this). Hence, I could agree that this manuscript is published in Nature communications, provided the editors consider the novelty sufficiently high.

Response: We thank the reviewer for the appreciation! Also, we are very grateful for your comments and suggestions that have significantly help us improve the overall quality of this manuscript.

Reviewer #3

Accept. The author answered all questions properly.

Response: We thank the reviewer for the appreciation of our work, for the insightful comments and for the useful suggestions, which have helped improve the overall quality of the manuscript.